# Opponent vesicular transporters regulate the strength of glutamatergic neurotransmission in a *C. elegans* sensory circuit

Jung-Hwan Choi [1], Lauren Bayer Horowitz[1] & Niels Ringstad [1✉]

At chemical synapses, neurotransmitters are packaged into synaptic vesicles that release their contents in response to depolarization. Despite its central role in synaptic function, regulation of the machinery that loads vesicles with neurotransmitters remains poorly understood. We find that synaptic glutamate signaling in a *C. elegans* chemosensory circuit is regulated by antagonistic interactions between the canonical vesicular glutamate transporter EAT-4/VGLUT and another vesicular transporter, VST-1. Loss of VST-1 strongly potentiates glutamate release from chemosensory BAG neurons and disrupts chemotaxis behavior. Analysis of the circuitry downstream of BAG neurons shows that excess glutamate release disrupts behavior by inappropriately recruiting RIA interneurons to the BAG-associated chemotaxis circuit. Our data indicate that in vivo the strength of glutamatergic synapses is controlled by regulation of neurotransmitter packaging into synaptic vesicles via functional coupling of VGLUT and VST-1.

[1] Department of Cell Biology, Skirball Institute of Biomolecular Medicine, Neuroscience Institute, NYU School of Medicine, New York, NY 10016, USA.
✉email: Niels.Ringstad@med.nyu.edu

Flow of information through neural circuits requires the regulated release of neurotransmitters at chemical synapses. The neurotransmitters that mediate synaptic signaling are stored in synaptic vesicles. These remarkable organelles accumulate neurotransmitters above their cytoplasmic concentrations and fuse with the plasma membrane upon neuronal depolarization to release their contents into the synaptic cleft. This fusion mechanism is common to neurons of all types[1,2]. By contrast, the mechanisms that package neurotransmitters into synaptic vesicles vary according to the neurotransmitter identity of a neuron. Each neurotransmitter is associated with a vesicular transporter that mediates its influx into the vesicular lumen[3]. Distinct transporters support the loading of synaptic vesicles with glutamate, GABA, acetylcholine, and monoamines such as serotonin and dopamine. Consequently, vesicular transporters are determinants of a key functional attribute of every neuron: what neurotransmitters that neuron uses to signal to synaptic partners.

Neurons can express more than one vesicular transporter and this not only determines which transmitters are released but also might determine how much of each transmitter is packaged into the synaptic vesicles. As all vesicular transmitters harness the energy stored in an electrochemical gradient generated by a vesicular proton pump (reviewed in ref. [4]), one vesicular transporter can influence the function of another. Vesicular glutamate transporter (VGLUT) 3 and vesicular acetylcholine transporter (VAChT) are co-expressed in striatal cholinergic interneurons, and glutamate transport through VGLUT3 potentiates acetylcholine transport by VAChT, by allowing the vesicular proton pump to generate a steeper pH gradient across the vesicle membrane[5]. Similarly, glutamate transport via VGLUT2, which is co-expressed in a subset of neurons with the vesicular monoamine transporter (VMAT), enhances VMAT-dependent dopamine transport into vesicles[6,7]. Based largely on biophysical studies, it has been suggested that interactions between vesicular transporters can regulate the strength of neurochemical signaling in specific circuits. The functional significance of interactions between vesicular transporters in neural circuits and on behavior, however, remains poorly understood. It is also likely that these interactions are widespread and involve additional vesicular transporters and channels, such as cation/H$^+$ exchangers and ClC chloride transporters/channels[8–11], which do not transport neurotransmitters but mediate the transport of other ions that can affect the electrochemical gradient. New vesicular transporters continue to be discovered[12,13], indicating that the current census of vesicular transporters is incomplete and raising the intriguing possibility that many neurotransmitter systems are regulated by interactions between vesicular transporters.

The nematode *Caenorhabditis elegans* is a powerful model for studying synaptic function in general and vesicular transporters in particular. Genetic and biochemical studies of *C. elegans* revealed the molecular identity of VAChT[14] and vesicular GABA transporter[15]. The characterization of the primary *C. elegans* VGLUT, EAT-4, and the assignment of its function to glutamatergic neurons in the pharyngeal nervous system helped establish the molecular identities of VGLUTs[16,17]. A particular strength of the *C. elegans* model is that simple and stereotyped behaviors critically depend on specific neurotransmitter signals. Genetic analysis of such behavior is a powerful method to identify molecular factors required for specific kinds of neurotransmission.

Synaptic glutamate signaling in *C. elegans* is required for a suite of chemosensory, mechanosensory, and thermosensory behaviors[18,19]. As in the vertebrate brain, glutamate is a major excitatory neurotransmitter in the *C. elegans* nervous system[16,20,21]. Of note, the molecular mechanisms of glutamatergic neurotransmission are highly conserved between *C. elegans*

and vertebrates. The *C. elegans* nervous system uses homologs of ionotropic AMPA receptors, kainate receptors, and *N*-methyl-D-aspartate receptors for fast, excitatory synaptic signaling[22–25]. Homologs of metabotropic glutamate receptors mediate G protein-coupled glutamate signaling in the *C. elegans* nervous system[26–29]. *C. elegans* neurons and glia use conserved mechanisms to package glutamate into synaptic vesicles and clear glutamate after its release, respectively[16,30,31].

Here we report the discovery of a vesicular transporter, VST-1, which is required in glutamatergic chemosensory neurons for chemotactic avoidance behavior. Loss of VST-1 causes a dramatic increase in the amount of glutamate released from $CO_2$-sensing neurons, which requires EAT-4/VGLUT. We find that excess glutamate signaling in *vst-1* mutants recruits interneurons to the chemotaxis circuit that are normally quiescent via AMPA-type glutamate receptors, and that this ectopic activation of interneurons in *vst-1* mutants is a major cause of their chemotaxis defects. These data show that a presynaptic mechanism that antagonizes VGLUT-dependent packaging of glutamate into synaptic vesicles is a critical determinant of the strength of synaptic glutamate signaling in vivo.

## Results

*C. elegans* possesses a pair of chemosensory neurons that detect the respiratory gas carbon dioxide ($CO_2$), the BAG neurons. When we analyzed their transcriptome for differentially expressed genes[32], we observed that BAG neurons are enriched for transcripts encoding EAT-4, the primary *C. elegans* VGLUT (Fig. 1a, c). We also noted that they are enriched for transcripts that encode the related transporter SLC-17.1 (Fig. 1b, c). As indicated by its name, SLC-17.1 is a member of the SLC17 family of solute transporters, which comprises the VGLUTs, the vesicular nucleotide transporter, sialin, and inorganic phosphate transporters[33] (Fig. 1d and Supplementary Fig. 1). For clarity, we hereafter refer to *slc-17.1* as *vst-1* (*vst* = vesicular solute transporter).

To determine whether VST-1 is required for the function of chemosensory BAG neurons, we tested whether mutation of *vst-1* or *vst-1* knockdown by RNA interference (RNAi) affected a chemotaxis behavior supported by BAGs. Under basal conditions, *C. elegans* avoid $CO_2$ and navigate down a $CO_2$ gradient[34,35]. As expected, wild-type animals placed in an arena with sectors containing either $CO_2$-enriched air or air with no $CO_2$ avoided the $CO_2$-enriched sector (Fig. 1e). Animals lacking the $CO_2$ receptor GCY-9[36,37] were profoundly defective in $CO_2$-avoidance and partitioned equally between the two sectors (Fig. 1e), consistent with prior studies[36,38]. Animals lacking EAT-4/VGLUT were also severely defective in $CO_2$ avoidance, indicating that glutamatergic signaling is essential for this behavior (Fig. 1e). Mutants carrying nonsense alleles of *vst-1* (Supplementary Fig. 2a) were also defective for $CO_2$ avoidance, as were *trans*-heterozygotes for different nonsense alleles (Fig. 1e). Furthermore, the $CO_2$ avoidance defect of *vst-1* mutants was rescued by a *vst-1* transgene (Fig. 1f and Supplementary Fig. 2b, c). These data show that VST-1 is required for BAG-dependent chemotaxis.

To test whether *vst-1* functions in BAG neurons, we expressed double-stranded RNA corresponding to *vst-1* coding sequences specifically in BAGs to trigger RNAi and knock down *vst-1*. BAG-specific RNAi targeting *vst-1* caused a $CO_2$-avoidance defect (Fig. 1e), indicating that *vst-1* is required in BAG neurons. The effect of *vst-1* knockdown in BAGs was comparable to that of knocking down *eat-4/VGLUT* in BAGs (Fig. 1e), indicating the functional importance of VST-1 in these glutamatergic sensory neurons.

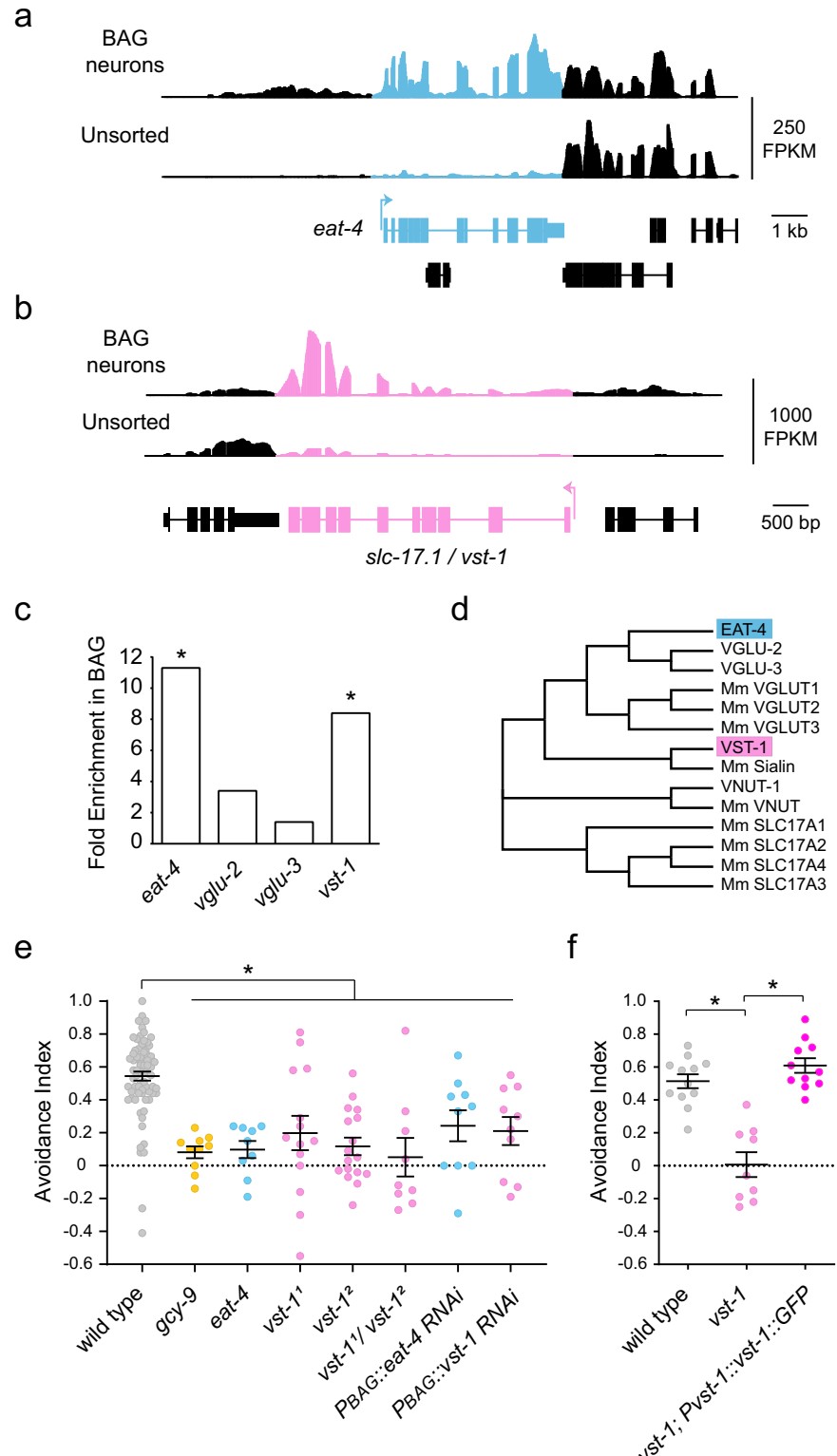

**VST-1 is a synaptic vesicle transporter expressed in a majority of glutamatergic neurons**. To determine which cells express *vst-1*, we generated a fosmid reporter that expresses nuclear mCherry in *vst-1*-expressing cells (Supplementary Fig. 2b). We made transgenic animals carrying this reporter together with an *eat-4* fosmid reporter that expresses nuclear yellow fluorescent protein (YFP) in glutamatergic neurons[21]. These reporters revealed that *vst-1* is expressed in many head neurons (Fig. 2a). Notably, a large majority of glutamatergic neurons in the head marked by the *eat-*

*4* reporter also expressed *vst-1* (Fig. 2b). We next asked where VST-1 protein localizes within the neurons. For this, we generated another *vst-1* reporter that encodes a functional VST-1::GFP fusion (Supplementary Fig. 2b, c). VST-1::GFP fluorescence was strikingly enriched in the nerve ring (Fig. 2c), a neuropil containing most of the synaptic connections in the *C. elegans* nervous system. Many synaptic proteins, including components of synaptic vesicles, are expressed in a similar pattern[14,15,39]. To test whether VST-1 is associated with synaptic vesicles, we introduced

**Fig. 1 vst-1 is required in BAG chemosensory neurons for $CO_2$ chemotaxis. a** Mean RNAseq signal (fragments per kilobase of transcript per million mapped read, FPKM) of *eat-4* obtained from two biological replicates. *eat-4* exons are blue and exons from neighboring genes are black. **b** Mean RNAseq signal (FPKM) of *vst-1* (*slc-17.1*) obtained from two biological replicates. *vst-1* exons are pink and exons from neighboring genes are black. **c** Fold enrichment in BAG neurons of *eat-4* transcripts and transcripts from *eat-4*-like genes. *False discovery rates of <0.01. **d** Dendrogram of a subset of *C. elegans* SLC17 family transporters related to VST-1 (species not indicated) and *M. musculus* SLC17 family transporters (indicated by Mm). SLC17A1-4 indicate the type I sodium-dependent inorganic phosphate transporters of the family. **e** Avoidance indices (mean ± SEM) of *vst-1* and *eat-4* mutants from measurements of $CO_2$-chemotaxis ($n = 74, 10, 9, 14, 17, 9, 10$, and 10 for wild type, *gcy-9*, *eat-4*, *vst-1¹*, *vst-1²*, *vst-1¹/vst-1²*, $P_{BAG}$::*eat-4* RNAi, and $P_{BAG}$::*vst-1* RNAi, respectively). *$P < 0.05$ calculated from comparisons of the mutants to wild type using a Kruskal–Wallis test corrected for multiple comparisons with Dunn's test (*P*-values for each comparison marked with an asterisk (*)—wild type vs. *gcy-9(tm2816)*: $5.5 \times 10^{-5}$; wild type vs. *eat-4(ky5)*: $3.0 \times 10^{-4}$; wild type vs. *vst-1¹*: 0.0040; wild type vs. *vst-1²*: $1.7 \times 10^{-6}$; wild type vs. *vst-1¹/vst-1²*: $1.7 \times 10^{-4}$; wild type vs. $P_{BAG}$::*eat-4* RNAi: 0.026; wild type vs. $P_{BAG}$::*vst-1* RNAi: 0.0094). For simplicity, *vst-1¹* was used to denote *vst-1(gk673717)* and *vst-1²* was used to denote *vst-1(gk308047)*. **f** $CO_2$ chemotaxis (mean avoidance indices ± SEM) of wild type ($n = 12$), *vst-1(gk308047)* ($n = 9$), and *vst-1(gk308047); Pvst-1::vst-1::GFP* ($n = 11$). *$P < 0.05$ calculated from comparisons to *vst-1* using a Kruskal–Wallis test corrected for multiple comparisons with Dunn's test (*P*-values for each comparison marked with an asterisk (*)—*vst-1* vs. wild type: 0.0014; *vst-1* vs. *vst-1(gk308047)*; *Pvst-1::vst-1::GFP*: $6.6 \times 10^{-5}$).

this VST-1::GFP reporter into *unc-104* mutants, which lack a kinesin required for the transport of synaptic vesicle components, including vesicular transporters, to synapses[15,40]. In *unc-104* mutants, VST::GFP was no longer highly enriched in the nerve ring and instead accumulated in neuronal cell bodies (Fig. 2c). These data provide evidence that VST-1 is present on synaptic vesicles. To further confirm that VST-1 is associated with synaptic vesicles, we prepared transgenic animals expressing VST-1::GFP and EAT-4::mCherry for immunogold staining and electron microscopy. Ultrathin sections of the nerve ring showed the circular profiles of fasciculated axons, many of which contained clusters of synaptic vesicles that were marked with anti-green fluorescent protein (GFP) immunogold (Fig. 2d). There was no immunogold labeling in the absence of anti-GFP primary antibody (Supplementary Fig. 3a), indicating that the immunogold label was specifically recognizing VST-1::GFP. We looked for colocalization of VST-1 and EAT-4 on the same synaptic vesicle by co-staining sections with 15 nm gold to detect VST-1::GFP and 5 nm gold to detect EAT-4::mCherry. Within presynaptic regions that contained both labels, we found synaptic vesicles that co-express VST-1 and EAT-4, i.e., vesicles associated with both large and small gold label (Supplementary Fig. 3b, c). Together, these data indicate that VST-1 is localized on synaptic vesicles and expressed by a majority of glutamatergic neurons in the *C. elegans* nervous system, and suggest that VST-1 can be co-expressed with EAT-4/VGLUT on synaptic vesicles.

**VST-1 inhibits EAT-4/VGLUT-dependent glutamate release from BAG neurons.** As VST-1 is an SLC17-family transporter related to EAT-4/VGLUT, we hypothesized that VST-1 and EAT-4 might function together to mediate glutamatergic neurotransmission. To test this hypothesis, we designed an assay to measure evoked glutamate release from BAG neurons. We expressed the glutamate sensor iGluSnFR[30] on the surface of BAG neurons, which we isolated in culture. Under these conditions, we could depolarize individual BAG neurons with high-potassium saline and measure changes in iGluSnFR fluorescence that reported glutamate release from that neuron. Wild-type BAG neurons displayed robust iGluSnFR signals upon depolarization (Fig. 3a). iGluSNFR signals were severely diminished by $CdCl_2$, a calcium channel inhibitor (Supplementary Fig. 4a, b), indicating that these signals reflected calcium-dependent release of glutamate from BAG neurons. Also, BAG neurons lacking EAT-4/VGLUT did not display any evoked iGluSnFR signals (Fig. 3b), further indicating that iGluSNFR signals report the evoked release of glutamate from BAG neurons. Of note, these experiments failed to reveal any EAT-4/VGLUT-independent mechanism for glutamate release from BAGs. We next determined the effect of *vst-1* mutation on glutamate release from BAGs. Surprisingly,

analysis of two independent *vst-1* mutants showed that loss of VST-1 significantly increased glutamate release from BAGs (Fig. 3c–f). We confirmed that mutation of either *vst-1* or *eat-4* had no marked effect on iGluSnFR expression in BAG neurons (Supplementary Fig. 4c). These data indicate that VST-1 antagonizes EAT-4/VGLUT-dependent glutamate release in BAG neurons.

We considered the possibility that VST-1 regulates glutamate release by controlling the amount of vesicle fusion elicited by synaptic depolarization, e.g., by regulating release probability or the size of the pool of vesicles competent for fusion. To test this, we expressed the exocytosis reporter synaptopHluorin[41] in cultured BAG neurons and asked whether *vst-1* mutation altered the amount of depolarization-evoked vesicle fusion. Wild-type and *vst-1* mutant BAG neurons displayed similar synaptopHluorin signals upon depolarization (Supplementary Fig. 4e, f), strongly suggesting that the increase in glutamate release caused by loss of VST-1 was not the result of increased vesicle fusion. *eat-4* mutant BAG neurons showed a similar increase in synaptopHluorin signal, suggesting vesicle fusion is intact in the absence of EAT-4/VGLUT (Supplementary Fig. 4e, f). As loss of VST-1 affected vesicular pH (see below), we adjusted synaptopHluorin signals for the observed effects of mutations on vesicular pH; even with this correction, we observed no effect of loss of vesicular transporters on vesicle exocytosis as measured by synaptopHluorin (Supplementary Fig. 4e', f'). Based on these data, we concluded that the vesicular transporter VST-1 regulates the amount of glutamate stored in synaptic vesicles, not the amount of vesicular fusion elicited by depolarization.

VGLUTs are members of a family of anion transporters that move diverse solutes, including inorganic phosphate, acidic sugars, negatively charged amino acids, and phosphorylated adenosine nucleotides[33]. As a member of the SLC17 family of transporters, VST-1 is likely an anion transporter and there are different ways an anion transporter in the synaptic vesicle membrane could limit glutamate uptake. VST-1 might compete with EAT-4/VGLUT for access to the electrochemical gradient used for glutamate import into synaptic vesicles. Alternatively, VST-1 might harness the same electrochemical gradient to drive glutamate efflux from synaptic vesicles. These different mechanisms would impact vesicular pH in different ways, suggesting an experimental approach to determine whether VST-1 mediates import of anions into the vesicle or instead functions as a glutamate efflux transporter. The former mechanism would increase the anion concentration in the synaptic vesicle and facilitate proton import by the vesicular $V_0$-ATPase, thereby acidifying synaptic vesicles. By contrast, a glutamate efflux transporter operating as other SLC17-type anion efflux transporters, e.g., Sialin/SLC17A5, is predicted to have the opposite effect

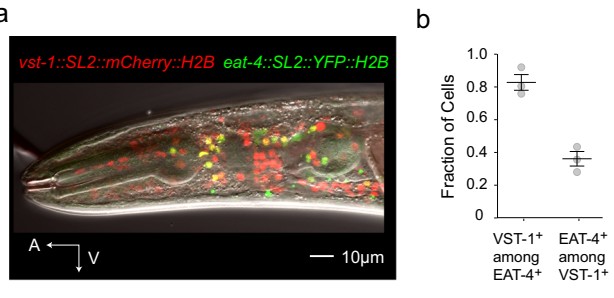

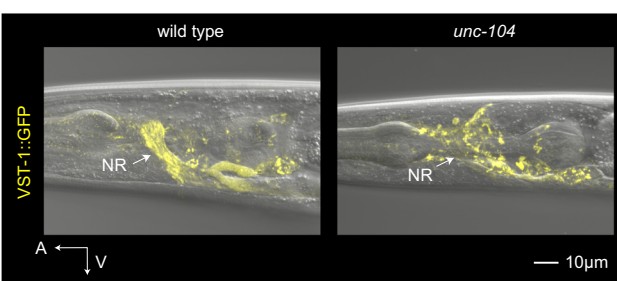

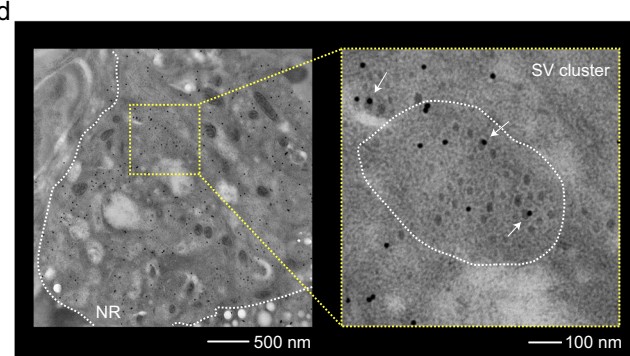

**Fig. 2 VST-1 is a synaptic vesicle transporter expressed by a majority of glutamatergic neurons. a** Micrograph of an animal expressing fosmid reporters of *eat-4/vglut* and *vst-1*. The *eat-4* and *vst-1* reporters express nuclear YFP (pseudocolored green) and nuclear mCherry (red), respectively. Anterior (A) and ventral (V) are indicated. **b** Quantification of *eat-4* and *vst-1* co-expression. The fraction (mean ± SEM) of *eat-4*-expressing cells also expressing *vst-1* and vice versa was determined from three independent experiments. **c** Localization of VST-1::GFP to presynaptic domains in the nervous system. The left panel shows a representative micrograph of a wild-type animal, in which VST-1::GFP is highly localized to the nerve ring (NR), marked with an arrow. The right panel shows VST-1::GFP expression in *unc-104/kinesin* mutants, which is required for proper transport of synaptic vesicle components into axons. The micrographs are representative of images taken from four animals for each genotype. **d** Immunoelectron micrograph of a strain expressing VST-1::GFP and stained with anti-GFP antibodies detected with 15 nm gold particles. The left panel shows a field of view at a magnification of ×25,000. The boundary of the nerve ring (NR) is marked by a dashed white line. The right panel shows a magnified view of the synaptic vesicle (SV)-rich region indicated by dashed yellow lines. The dashed white line in the right panel indicates the perimeter of a single neurite cut in cross-section. Immunogold particles (15 nm) show the expression of VST-1::GFP. Arrows indicate immunogold associated with synaptic vesicles. The micrograph is representative of at least 25 micrographs taken of different regions of the nerve ring.

on vesicular pH. To distinguish these possibilities, we used synaptopHluorin to measure vesicular pH in wild-type and *vst-1* BAG neurons. Measurements of total and surface-accessible pHluorin (Fig. 3g) allow computation of vesicular pH[42]. We observed that wild-type synaptic vesicles had a mean pH of 6.2

(Fig. 3h). This is higher than the vesicular pH measured in cultured hippocampal neurons from mammals, which is estimated to be between 5.6 and 5.8[41,43–45]. The reason for this difference is not clear, but we note that vesicular pH has only been measured in a few neuron types and the extent to which vesicular pH varies between different kinds of glutamatergic neurons within the mammalian nervous system is not known.

Importantly, we found that loss of VST-1 caused a measurable increase in vesicular pH (Fig. 3h), consistent with a model in which VST-1 supports anion influx into synaptic vesicles. We also measured vesicular pH in BAG neurons lacking EAT-4/VGLUT (Fig. 3h). Unlike loss of VST-1, loss of EAT-4/VGLUT did not cause a measurable change in vesicular pH. The effect of VST-1 mutation on vesicular pH provides additional evidence that VST-1 functions in the synaptic vesicle membrane. These data are also consistent with a model in which VST-1 is an anion transporter that competes with EAT-4/VGLUT for the electrochemical gradient required for glutamate uptake into synaptic vesicles. However, some SLC17 family transporters can cotransport cations such as $Na^+$ and $H^+$ [33], and we cannot rule out the possibility that cation efflux (rather than anion influx) contributes to the effect of VST-1 on vesicular pH.

**Behavioral defects of *vst-1* mutants are caused by excess signaling through AMPA-type glutamate receptors.** Loss of VST-1 markedly increases evoked glutamate release from chemosensory BAG neurons. Does excess glutamate release cause the chemotaxis defects of *vst-1* mutants? We reasoned that, if so, attenuating postsynaptic glutamate receptors might restore chemotaxis to *vst-1* mutants (Fig. 4a). The *C. elegans* genome encodes metabotropic and ionotropic glutamate receptors[20,26,46]. We tested our hypothesis by targeting GLR-1, an AMPA-type receptor that is expressed by many neurons postsynaptic to BAGs[20,47]. GLR-1 receptors also function in circuits that process inputs from glutamatergic sensory neurons[24,48].

When we tested whether GLR-1 receptors were required for BAG-mediated chemotaxis, we found that *glr-1* mutants were able to avoid $CO_2$ nearly as well as the wild type (Fig. 4b), indicating that GLR-1 receptors are dispensable for the chemotaxis circuit downstream of BAGs. Strikingly, loss of GLR-1 receptors restored chemotaxis behavior to *vst-1* mutants (Fig. 4b), consistent with a model in which loss of VST-1 causes chemotaxis defects via excess activation of glutamate receptors. By contrast, loss of EAT-4/VGLUT did not restore $CO_2$ avoidance to *vst-1* mutants, confirming the essential role of glutamate signaling in BAG-mediated avoidance behavior. Together, these data indicate that exuberant glutamate release by *vst-1* mutant neurons causes behavior defects in vivo through excess activation of AMPA-type receptors.

To identify specific motor programs that are impacted by excess glutamate signaling in *vst-1* mutants, we used high-resolution videotracking to measure acute behavioral responses to $CO_2$ stimuli of the wild type, as well as *vst-1* and *glr-1* mutants. Animals in a chamber were exposed to pulses of $CO_2$-enriched atmosphere, while their locomotion was recorded. From the recorded trajectories, we computed linear speed and instantaneous frequency of high-angle turns. Each of these parameters changed during a wild-type response to $CO_2$ stimuli: upon sensing $CO_2$, the speed of foraging animals decreased and their turn-frequency increased (Fig. 5a, d). Slowing and turning responses required the BAG-specific $CO_2$ receptor GCY-9 (Supplementary Fig. 5). We observed that mutation of *vst-1* affected parameters of foraging behavior prior to the $CO_2$ stimulus, i.e., basal foraging behavior, as well as parameters of $CO_2$-evoked behavior. Basal and evoked speeds of *vst-1* mutants

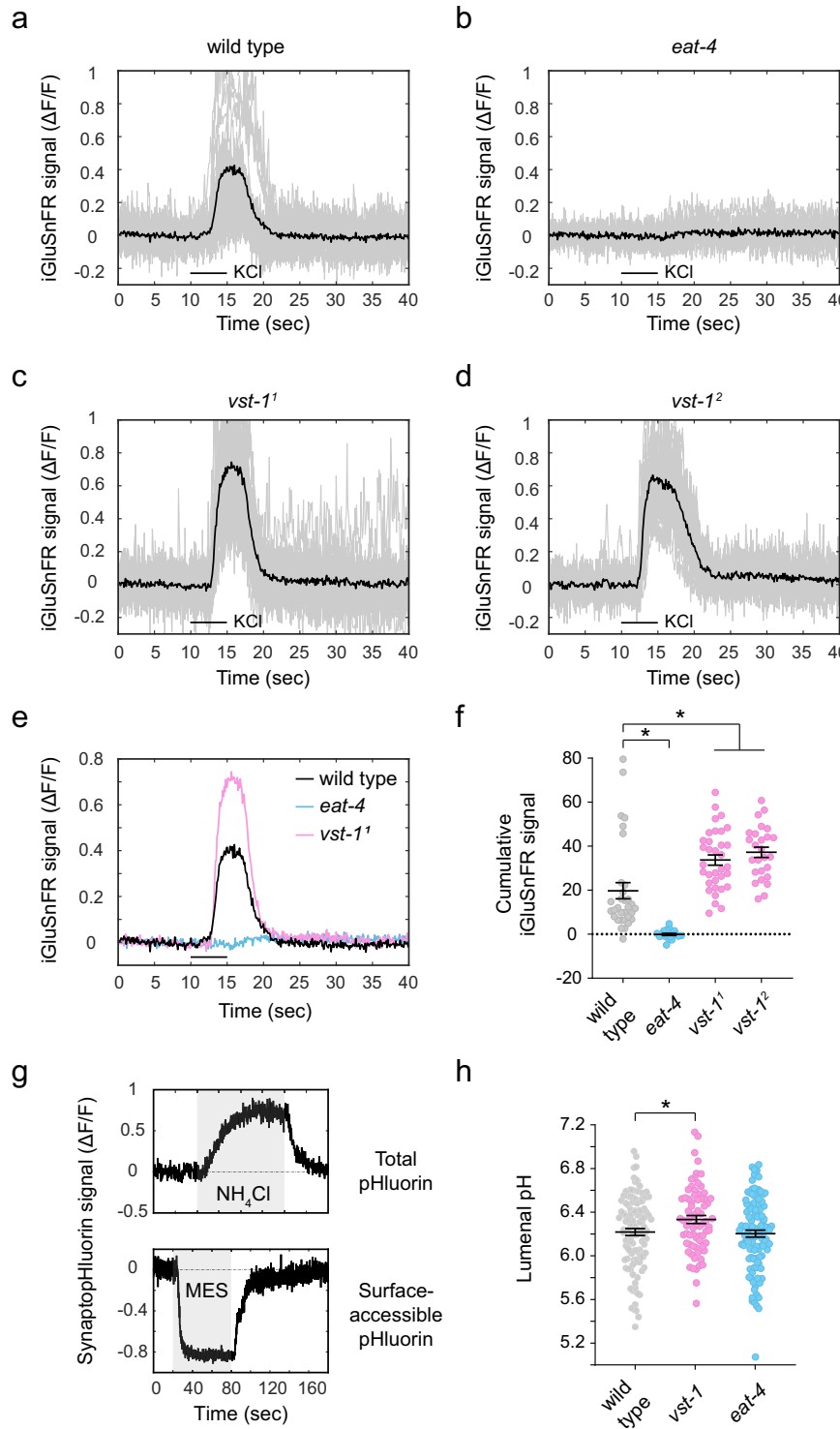

were significantly lower than those of the wild type (Fig. 5a, d). Also, after repeated stimulation, the evoked turning response of *vst-1* mutants was significantly greater than that of the wild type, suggesting that these mutants became partially sensitized to $CO_2$ stimuli in a manner that the wild type did not (Fig. 5a, d). These data show that VST-1 is required for normal foraging behavior and for acute behavioral responses to $CO_2$ stimuli.

Importantly, many effects of *vst-1* mutation depended upon synaptic glutamate signaling. *eat-4/VGLUT* mutants displayed abnormal slowing responses to $CO_2$ stimuli and their turning responses to repeated $CO_2$ stimuli also differed from those of the

wild type by partially adapting (Fig. 5b, d). Mutation of *vst-1* in *eat-4* mutants had no further effect on basal speed or on $CO_2$-evoked slowing (Fig. 5b, d). We did observe a small but significant effect of *vst-1* mutation on $CO_2$-evoked turning of *eat-4* mutants (Fig. 5b, d), but in *eat-4* mutants, loss of VST-1 caused a decrease in the frequency of turning, unlike in the wild type, whose evoked turning was greatly increased by loss of VST-1. Mutants lacking the AMPA-type receptor GLR-1 displayed grossly normal basal and $CO_2$-evoked behavior. Importantly, loss of VST-1 had no significant effect on the behavior of *glr-1* mutants (Fig. 5c, d). These data indicate that the locomotory defects of *vst-1* mutants

**Fig. 3 VST-1 inhibits EAT-4/VGLUT-dependent glutamate release from BAG neurons and acidifies synaptic vesicles. a–d** iGluSnFR signal ($\Delta F/F$) from neurites of BAG neurons in culture expressing the glutamate sensor iGluSnFR. Individual traces are plotted in gray and the mean signal is plotted in black. Neurons were depolarized with a five second pulse of 100 mM KCl. BAG neurons were isolated from the **a** wild type ($n = 32$), **b** eat-4(ky5) mutants ($n = 19$), **c** vst-1(gk673717) mutants (vst-1$^1$) ($n = 34$), and **d** vst-1(gk308047) mutants (vst-1$^2$) ($n = 26$). **e** Mean iGluSnFR signals from wild-type, eat-4, and vst-1 BAG neurons. Line under traces indicates KCl application. The overlay shows increased glutamate release from vst-1 neurons and no glutamate release from eat-4 neurons. **f** Quantification of iGluSnFR signals (mean ± SEM) from wild-type and mutant BAG neurons shown in **a–d**. Cumulative signal between 10 and 20 s was computed for each trial. *$P < 0.05$ for comparisons of the mutants to wild type as determined by a Kruskal–Wallis test and corrected for multiple comparisons via Dunn's test (P-values for each comparison marked with an asterisk (*)—wild type vs. eat-4: 0.0004; wild type vs. vst-1$^1$: 0.0035; wild type vs. vst-1$^2$: 0.0006). **g** Measurement of total synaptopHluorin and surface-accessible synaptopHluorin used to compute vesicular pH. Change of synaptopHluorin signal ($\Delta F/F$) of an example KCl-responsive punctum in response to NH$_4$Cl and MES over time is shown. These measurements were used to compute lumenal pH according to Mitchell et al.[68]. **h** Synaptic vesicle pH (mean ± SEM) of wild type, vst-1$^2$ mutants, and eat-4 mutant BAG neurons ($n = 112$, 71, and 110, respectively). *$P < 0.05$ for comparisons of the mutants to wild type as determined by one-way ANOVA corrected for multiple comparisons via Dunnett's test (the pH of all genotypes have a normal distribution, P-value for each comparison—wild type vs. vst-1$^2$: 0.041 wild type vs. eat-4: 0.93).

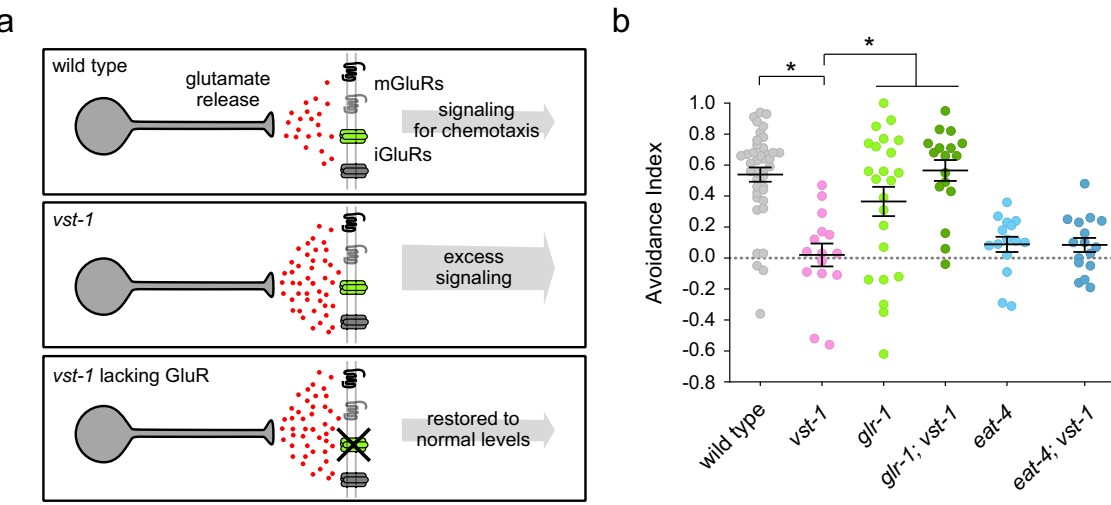

**Fig. 4 The CO$_2$ chemotaxis defect of vst-1 mutants requires an AMPA-type glutamate receptor. a** Schematic of a test for the hypothesis that excess glutamate signaling causes chemotaxis defects by over-activating glutamate receptors. mGluRs: metabotrobic glutamate receptors; iGluRs: ionotropic glutamate receptors. **b** Avoidance indices (mean ± SEM) of vst-1(gk308047), glr-1(n2461), and eat-4(ky5) mutants during chemotaxis from 10% CO$_2$ ($n = 40$, 15, 23, 17, 15, and 16 for wild type, vst-1, glr-1, glr-1;vst-1, eat-4, and eat-4;vst-1, respectively). *$P < 0.05$ from comparisons of the indicated strains to vst-1 mutants as determined by a Kruskal–Wallis test corrected for multiple comparisons via Dunn's test (P-values for each comparison marked with an asterisk (*)—vst-1$^2$ vs. wild type: $3.5 \times 10^{-5}$; vst-1$^2$ vs. glr-1: 0.019; vst-1$^2$ vs. glr-1;vst-1$^2$: $2.0 \times 10^{-4}$).

was suppressed by loss of GLR-1 (Fig. 5c, d). Together, these data indicate that the chemotaxis defects of vst-1 mutants and their defects in basal locomotion and acute responses to CO$_2$ are likely caused by inappropriate activation of AMPA-type glutamate receptors.

**Loss of VST-1 increases AMPAR-dependent synaptic glutamate signaling from BAG sensory neurons to RIA interneurons.** Are there specific cells in the chemotaxis circuit downstream of BAG neurons that are affected by dysregulated glutamate signaling in vst-1 mutants? To answer this, we measured physiological responses to CO$_2$ stimuli of interneurons postsynaptic to BAGs. As the effects of vst-1 on CO$_2$-evoked behavior require GLR-1 glutamate receptors, we focused our studies on the subset of BAG targets that express GLR-1[20,47], which includes AIB, RIA, AVA, and AVE interneurons (Fig. 6a). We used a microfluidic device to deliver CO$_2$ stimuli to immobilized animals, while recording neuronal calcium responses. Under the conditions that we used, CO$_2$ stimuli elicited rapid and robust responses of BAG neurons (Supplementary Fig. 6a). During stimulus presentation, wild-type and vst-1 mutant BAGs showed similar responses to CO$_2$ stimuli (Supplementary Fig. 6a,

b), suggesting that loss of VST-1 did not grossly affect chemo-transduction in BAGs. We noted that recovery of BAG calcium to baseline was slower in vst-1 mutants, although we could not rule out the possibility that this difference was the result of experimental and sampling error ($P = 0.051$) (Supplementary Fig. 6c). This effect was not observed until later in the experiment and was possibly the result of altered feedback from downstream circuitry onto BAGs in vst-1 mutants. For this reason, we quantified interneuron activity during the time of stimulus presentation, when vst-1 mutation had no measurable effect on BAG neuron activity.

There were interneurons that displayed responses to BAG activation unaffected or only modestly affected by vst-1 mutation. Loss of VST-1 did not affect the calcium responses of AVA interneurons, which frequently decreased after BAG activation (Supplementary Fig. 7a). These data suggest that AVA interneurons receive inhibition from BAGs but that this inhibition is not affected by loss of VST-1. By contrast, AIY interneurons displayed increases in cell calcium in response to BAG activation, indicating a strong excitatory connection between BAGs and AIYs. However, AIY responses were only slightly increased by loss of VST-1 and we could not rule out the possibility that the

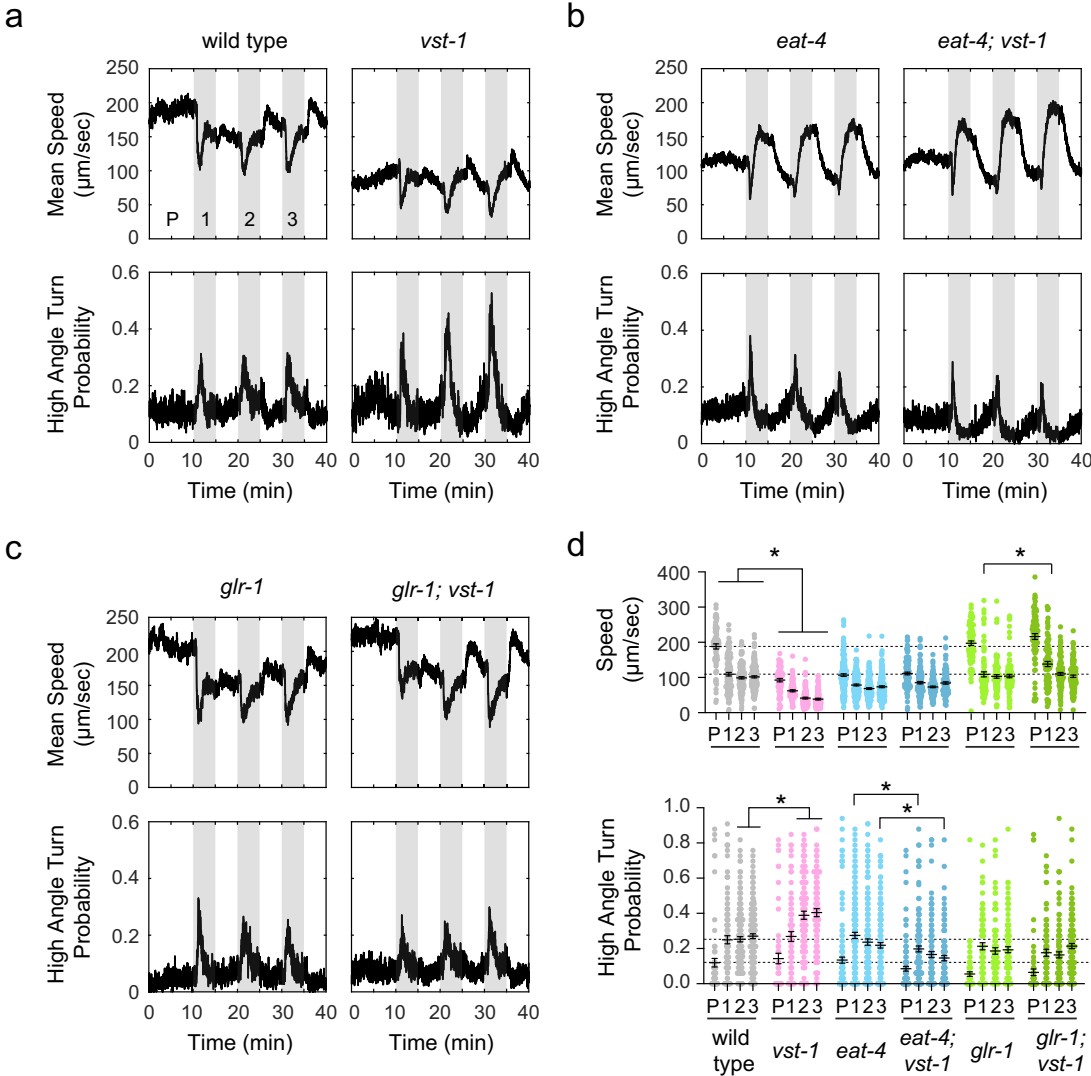

**Fig. 5 Loss of VST-1 disrupts acute behavioral responses to $CO_2$ in a manner than requires GLR-1/AMPAR and EAT-4/VGLUT. a** Instantaneous speed and probabilities of executing a high-angle turn of wild-type and *vst-1* animals in response to three pulses of $CO_2$. Pulses of $CO_2$ are labeled numerically and the pre-stimulus period is labeled 'P.' Traces in this panel and all other panels show the mean of measurements taken from at least eight independent trials of 30–60 animals. **b** Mean instantaneous speed and turn probabilities of *eat-4* and *eat-4; vst-1* mutants. **c** Mean instantaneous speed and turn probabilities of *glr-1* and *glr-1; vst-1* mutants. **d** Summary data from experiments shown in **a**–**c**. Dot plots show mean speeds and turn probabilities ± SEM of different strains during the pre-stimulus period (P) and after each presentation of $CO_2$ (labeled 1–3). The number of tracks analyzed (*n*) for mean speed during each period (P, 1, 2, and 3) are: 74, 105, 144, and 159 (wild type); 62, 79, 98, and 90 (*vst-1*); 147, 166, 151, and 163 (*eat-4*); 115, 130, 121, and 110 (*eat-4; vst-1*); 73, 85, 90, and 95 (*glr-1*); and 79, 84, 99, and 131 (*glr-1; vst-1*). The number of tracks analyzed (*n*) for high-angle turn probability during each period (P, 1, 2, and 3) are: 74, 98, 145, and 158 (wild type); 62, 72, 100, and 90 (*vst-1*); 147, 166, 155, and 170 (*eat-4*); 115, 131, 120, and 107 (*eat-4; vst-1*); 73, 85, 93, and 97 (*glr-1*); and 79, 84, 102, and 127 (*glr-1; vst-1*). Dashed lines indicate the means of wild type during periods P and 1. *$P < 0.05$ for comparisons between genotypes during the same period determined by a Kruskal–Wallis test and corrected for multiple comparisons via Dunn's test. Only statistical comparisons between genotypes shown in the same panel in **a**–**c** are indicated here for simplicity. Statistical comparisons between genotypes shown in different panels (e.g., comparison between *vst-1* and *eat-4; vst-1*) and exact *P*-values for all comparisons are shown in Supplementary Table 1.

observed differences resulted from experimental and sampling errors (Supplementary Fig. 7b).

Connections between BAGs and other interneurons were affected by VST-1 mutation. In the wild type, AIBs responded variably to BAG activation with either increases or decreases in cell calcium; on average, these responses were balanced and the mean response of AIBs to BAG activation was close to zero (Supplementary Fig. 7c). In *vst-1* mutants, AIBs displayed a decrease in cell calcium in response to BAG activation (Supplementary Fig. 7c), suggesting that loss of VST-1 augmented an inhibitory signal from BAGs to AIBs. Similar to AIB neurons, wild-type RIA neurons displayed variable responses to BAG

activation and the average response was close to zero (Fig. 6b, e). In *vst-1* mutants, RIAs exhibited an increase in cell calcium in response to BAG activation (Fig. 6b, e), suggesting that loss of VST-1 potentiated an excitatory signal from BAGs to RIAs. As the AMPA-type receptor GLR-1 is required for the chemotaxis defect of VST-1 and mediates excitatory glutamate signaling, we further investigated the role of RIA interneurons in mediating the *glr-1*-dependent behavioral response to $CO_2$ in *vst-1* mutants.

To demonstrate that the effect of *vst-1* mutation on RIA was the result of potentiated signaling from BAGs, we measured RIA responses to $CO_2$ stimuli in *gcy-9* mutants, which lack a key component of the $CO_2$-sensing machinery in BAGs[36,37]. Loss of

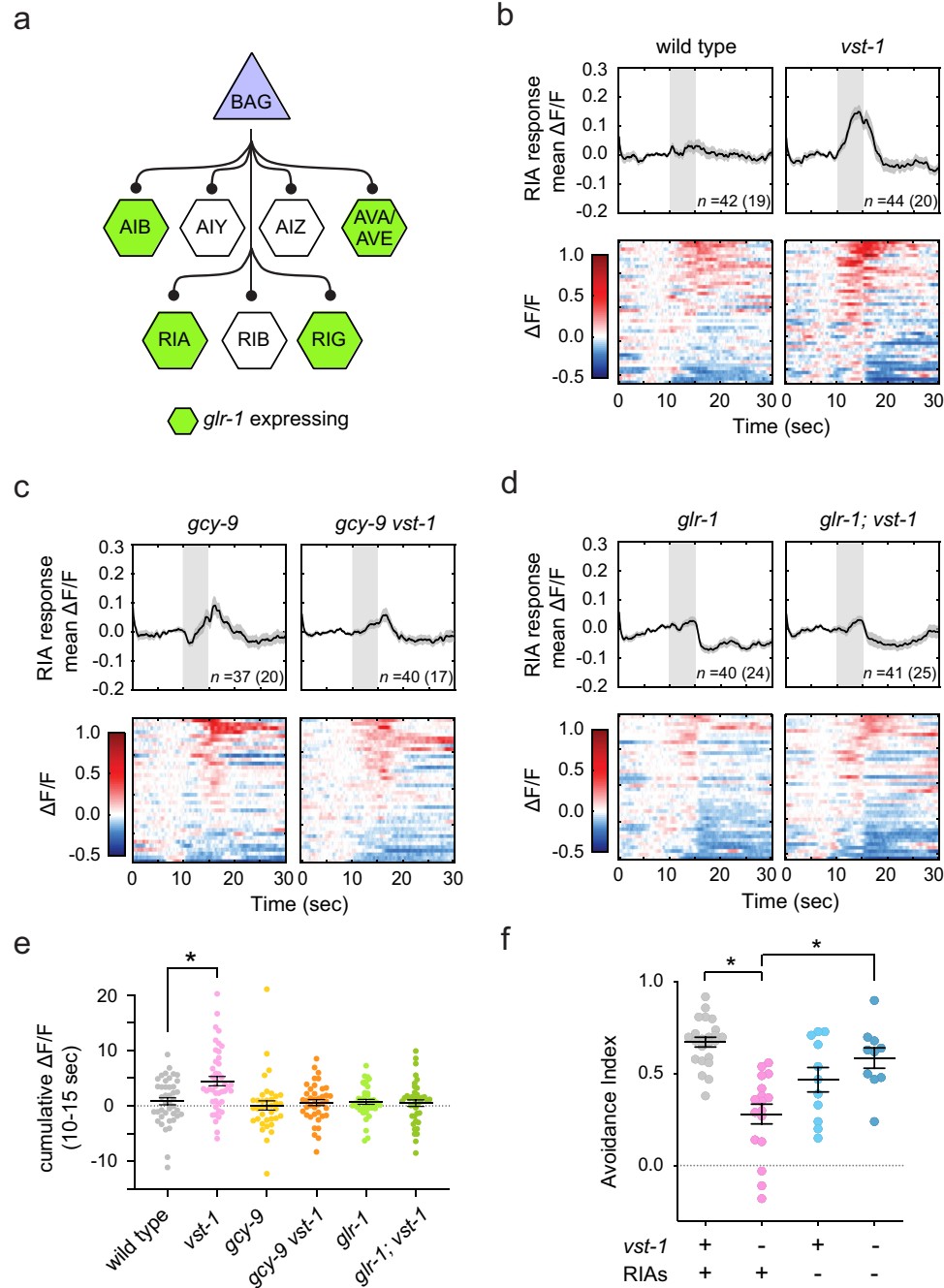

**Fig. 6 Loss of VST-1 alters the chemotaxis circuit downstream of BAGs by engaging RIA interneurons. a** A schematic showing interneurons that receive synaptic input from BAG sensory neurons. **b**–**d** GCaMP signal ($\Delta F/F$) measured in RIA neurons (*Pglr-3a::GCaMP3*) during stimulation of BAGs with 10% $CO_2$ stimuli. Upper panels show the mean signal ± SEM as black lines and gray shaded regions, respectively. Individual traces were passed through a 0.5 s moving-average filter for the plots (raw traces were used for quantification of GCaMP signal in **e**). Lower panels show individual responses as colormaps. The number in parentheses indicates the number of worms analyzed and *n* indicates the total number of cells analyzed. **e** Summary data from experiments shown in **b**–**d**. The dot plots show cumulative GCaMP signal (mean ± SEM) during $CO_2$ presentation (10–15 s) of the indicated genotypes. *$P < 0.05$ determined by a Kruskal–Wallis test and corrected for multiple comparisons between wild type and other genotypes via Dunn's test (*P*-value for wild type vs. *vst-1*: 0.017). **f** Effects of RIA ablation on $CO_2$ chemotaxis of the wild type and *vst-1* mutants. The mean ± SEM of avoidance indices for each condition are indicated (*n* = 23, 17, 11, and 10 for wild type, *vst-1*, RIA ablation in wild-type background, and RIA ablation in *vst-1* mutants, respectively). *$P < 0.05$ determined by a Kruskal–Wallis test and corrected for multiple comparisons between *vst-1* mutants and other genotypes via Dunn's test (*P*-values for each comparison—*vst-1* vs. wild type: $8.7 \times 10^{-7}$; *vst-1* vs. RIA ablation in wild-type background: 0.17; *vst-1* vs. RIA ablation in *vst-1* mutants: 0.014).

GCY-9 in *vst-1* mutants suppressed the effect of *vst-1* mutation, suggesting the BAG $CO_2$-sensing machinery is required for its effect (Fig. 6c, e). We further tested whether the effects of *vst-1* mutation on RIA activation by BAGs require GLR-1 glutamate receptors, as predicted by our model. In mutants lacking GLR-1,

there was no clear effect of *vst-1* mutation (Fig. 6d, e), indicating that the increased activation of RIAs observed in *vst-1* mutants requires signaling through GLR-1.

To determine whether increased activation of RIA contributes to the chemotaxis defects of *vst-1* mutants, we tested the effect of

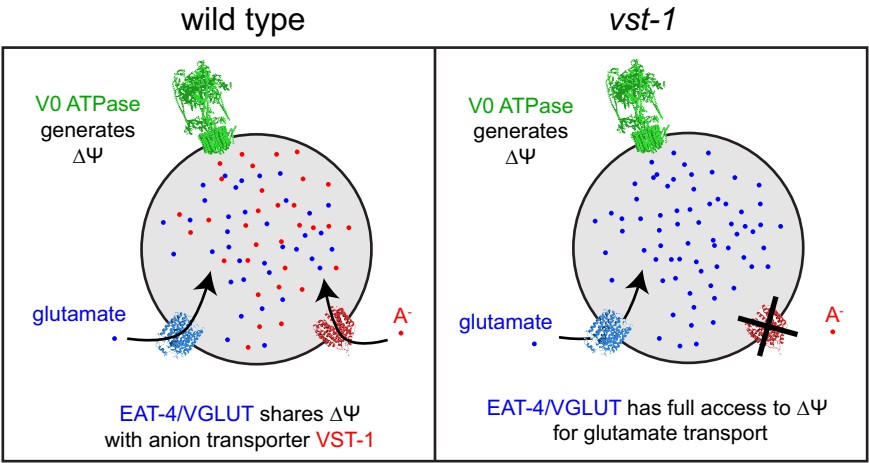

**Fig. 7 A model for VST-1 regulation of glutamate transport into synaptic vesicles.** A transmembrane potential ΔΨ generated by the vesicular ATPase is harnessed by VGLUT to transport glutamate (illustrated as blue dots) into synaptic vesicles. In our model, VST-1-mediated transport of an anion (A⁻, red dots) into the vesicle dissipates ΔΨ and thereby reduces glutamate transport.

*vst-1* mutation on $CO_2$ avoidance in animals lacking RIA interneurons. As we observed previously, *vst-1* mutants displayed a robust chemotaxis defect (Fig. 6f). Animals lacking RIAs in a wild-type background did not display a clear defect in chemotaxis behavior (Fig. 6f). Notably, loss of RIAs in *vst-1* mutants restored $CO_2$ chemotaxis (Fig. 6f). These data indicate that the chemotaxis defect of *vst-1* mutants is caused, at least in part, by ectopic activation of RIA interneurons.

## Discussion

**A presynaptic mechanism sets the strength of glutamatergic synapses via regulation of neurotransmitter loading.** Proper regulation of synaptic strength is critical for the function of neural circuits and mechanisms that change synaptic strength are drivers of circuit plasticity and are required for adaptation and learning. Well-understood plasticity mechanisms remodel or modulate the complement of postsynaptic neurotransmitter receptors to alter how synaptic signals are processed by the receiving neuron. Such mechanisms play critical roles in glutamatergic synapses and their molecular underpinnings have been extensively studied (reviewed in ref. [49]). Other plasticity mechanisms regulate the kinetics of synaptic vesicle fusion, either by altering presynaptic calcium levels or by modulating the fusion machinery itself (reviewed in ref. [50]). Our study of a chemotaxis circuit in *C. elegans* suggests that the strength of glutamatergic synapses can also be determined by a mechanism that sets the amount of glutamate packaged into synaptic vesicles. We find that the amount of glutamate released by chemosensory neurons is controlled by interactions between VGLUT and another vesicular transporter—VST-1— that opposes VGLUT function. Increasing glutamatergic neurotransmission by disrupting the balanced antagonism between VGLUT and VST-1 causes a behavioral defect that was as severe as that caused by mutations that eliminate glutamatergic neurotransmission, e.g., VGLUT mutation. This observation demonstrates the importance of mechanisms that assign appropriate strengths to functional connections within the chemotaxis circuit downstream of BAG neurons.

How does VST-1 oppose the function of VGLUT to determine the strength of glutamatergic signaling? Functional interactions between these transporters must be considered in the context of what is known about the energetics of neurotransmitter transport. It is possible that VST-1 antagonizes VGLUT by driving glutamate export from vesicles and directly counteracting VGLUT-mediated glutamate import. This "revolving-door"

model for glutamate transport across synaptic vesicle membranes demands that VST-1 be a glutamate efflux transporter. We do not favor this model for two reasons. First, it seems highly unlikely that VST-1 is a glutamate transporter. Recent structural studies of VGLUT have identified key residues implicated in glutamate binding[51] and these residues are not conserved between VST-1 and VGLUT (Supplementary Fig. 8). Second, our analysis of the effects of VST-1 mutation on vesicular pH indicated that loss of VST-1 alkalinizes vesicles, which is consistent with VST-1 mediating anion influx or cation efflux, not glutamate efflux. We therefore favor a model in which VST-1 competes with VGLUT for access to ΔΨ, the electrical potential gradient across the vesicular membrane (Fig. 7). An important implication of this model is that it predicts that the abundance of VST-1 substrates would regulate the glutamate content of synaptic vesicles and synaptic transmission in a chemotaxis circuit. Such substrates would, therefore, function as endogenous small-molecule regulators of glutamatergic neurotransmission. What this substrate is, how the concentration of this substrate is controlled, and whether its transport is coupled to other ions that can affect the electrical and osmotic balance of synaptic vesicles[52] are important questions that remain to be addressed in order to fully understand the role of VST-1 at synaptic vesicles.

**The strength of synaptic glutamate signaling is tuned to support BAG-dependent chemotaxis.** Loss of VST-1 causes a chemotaxis defect that is as severe as that caused by loss of VGLUT. This observation indicates that the strengths of glutamatergic synapses in the chemotaxis circuitry must be set within a certain range in order to correctly process inputs from BAG sensory neurons. BAG-dependent chemotaxis, such as other chemotaxis behaviors[53–55], is the result of a sensory system coordinating changes to motor programs that control speed and turning. We found that decreased and increased glutamate signaling caused by loss of VGLUT and loss of VST-1, respectively, did not eliminate the behavioral components of chemotaxis but instead disrupted their coordination. Loss of glutamate signaling had a dramatic effect on speed control and resulted in animals that were incapable of sustained slowing during BAG stimulation. By contrast, increased glutamate signaling caused by *vst-1* mutation resulted in persistent and stimulus-independent slowing. *vst-1* mutants also displayed exaggerated turning responses upon BAG stimulation, an effect that was not anticipated by our measurements of turning by *eat-4/VGLUT* mutants. It is noteworthy that the

behavior defects caused by loss of EAT-4/VGLUT and VST-1 are not precise mirror images of each other. This might indicate that loss of VST-1 does not affect all glutamatergic synapses equally, unlike loss of EAT-4/VGLUT, which is required across the board for all types of glutamatergic neurotransmission. Importantly, our data indicate that the effects of *vst-1* mutation on behavior require the AMPA-type glutamate receptor GLR-1 and are occluded by the effects of *eat-4/VGLUT* mutation, strongly indicating that VST-1 dysregulates synaptic glutamate signaling in vivo.

Our analysis of neurons that receive synaptic inputs from BAGs might also suggest that synapses in the chemotaxis circuit are differently sensitive to loss of VST-1. We found that BAG signaling to the RIA and AIB interneurons was significantly affected by loss of VST-1, whereas signaling from BAGs to other targets was either not affected or only affected slightly (Fig. 6b and Supplementary Fig. 7). In *vst-1* mutants, RIAs, on average, received more excitatory input from BAGs (Fig. 6b) and this excitation required the AMPA-type glutamate receptor GLR-1 (Fig. 6d), which is a critical mediator of fast excitatory signaling in the *C. elegans* nervous system[22–24]. On the other hand, AIBs received more inhibitory input on average in *vst-1* mutants. These observations might suggest that VST-1 has a privileged role at certain synapses in the chemotaxis circuit. At this time, we do not know the molecular basis for *vst-1* mutation preferentially affecting certain synapses. Future studies into potential modulatory mechanisms that synergize with glutamatergic transmission, excitatory/inhibitory receptor localization relative to release site, or the excitatory/inhibitory input of other neurons in the circuit could shed light on this question.

**Interactions between vesicular transporters as a mechanism for regulating neurotransmission**. We suggest that the functional interaction between VGLUT and VST-1 in glutamatergic synapses is an example of a more general mechanism that regulates neurotransmission by controlling neurotransmitter packing into synaptic vesicles. There is evidence for such a mechanism from prior studies of monoaminergic and cholinergic neurotransmission. Dopaminergic neurons in the vertebrate and insect brain co-express VMAT and VGLUT[6,7]. Eliminating VGLUT from dopaminergic neurons reduces dopamine storage in presynaptic terminals and diminishes dopamine release. VGLUT facilitates dopamine storage by facilitating the energetics of VMAT-dependent dopamine transport. VMAT relies on a proton gradient to transport dopamine into vesicles and vesicular glutamate offsets the charge imbalance caused by proton transport permitting a steeper proton gradient and, as a consequence, more dopamine transport. A similar interaction between VGLUT and VAChT has been proposed to explain how VGLUT boosts cholinergic signaling in the vertebrate striatum[5]. VGLUT-VMAT and VGLUT-VAChT interactions illustrate how vesicular transporters that access different components of the electrochemical gradient that powers neurotransmitter uptake can mutually reinforce their distinct transport functions.

Besides these examples, how widespread are functional interactions between synaptic vesicular transporters? This is a difficult question to answer, in no small part because the full complement of synaptic vesicle-associated transporters remains to be determined. There is no sequence-based hallmark of vesicular transporters and these factors are expressed at low levels, which confounds biochemical approaches for their identification. New vesicular transporters continue to be discovered; in addition to VST-1, recent studies of neural circuits in the insect brain have identified a transporter named Portabella[12], which is related to VMAT and VAChT, and a transporter named LOVIT, which is expressed in histaminergic

neurons[13]. The continued characterization of the large and diverse family of transporter proteins will likely add to the census of transporters in synaptic vesicle membranes and each of these transporters is, in principle, capable of modulating neurotransmission via the mechanisms discussed.

With respect to VST-1, our data suggest that this particular transporter has functions in non-glutamatergic neurons in addition to its function in glutamatergic neurons. VST-1 is clearly expressed by many non-glutamatergic neurons (Fig. 2a), raising the possibility that VST-1, such as vertebrate VGLUT isoforms, might potentiate neurotransmitter packaging mechanisms that require proton gradients, e.g., monoamine and acetylcholine transport. Therefore, depending on the energetics of the neurotransmitter transporters in a synaptic vesicle membrane, the function of VST-1 might change from antagonizing neurotransmitter uptake, as it does glutamate uptake, to potentiating it. In a *C. elegans* chemotaxis circuit, VST-1 is required as an auxiliary transporter that regulates VGLUT-dependent glutamate signaling. Such auxiliary transporters might be widespread regulators of synaptic neurotransmission in other circuits and it is intriguing to consider the possibility that their functional interactions with neurotransmitter transporters constitute mechanisms that generally regulate synaptic strength in vivo.

## Methods

**C. elegans strains**. Mutant and transgenic animals used for this study are listed in Supplementary Table 2. The *Pvst-1::vst-1::GFP* transgenic used for Fig. 1f (FQ1117) was introduced by microparticle bombardment of pJC75 (*Pvst-1::vst-1::GFP* fosmid) and co-transfection markers according to Praitis et al.[56], and subsequently crossed into the *vst-1(gk308047)* background. All other transgenic animals used for experiments were generated by microinjection following the standard protocol[57]. For most lines generated by injection of a plasmid, the plasmid was injected at 20–40 ng/μl concentration. The exceptions were:

 RNAi plasmids (pJC141, pJC143, pJC133, pJC135): 100 ng/μl

 pJC151 (*Pgcy-33::snb-1::superecliptic pHluorin*): 5 ng/μl

 *Punc-122::GFP, Punc-122::mCherry*: 60 ng/μl

For the *vst-1* fosmid reporter line (FQ1571), a linearized *vst-1* fosmid reporter (pEH62) was injected at 11 ng/μl concentration into a *eat-4* fosmid reporter line (OH11124; *otIs388[eat-4 fosmid::SL2::YFP::H2B + (pBX) pha-1(+)] pha-1(e2123)*) with carrier DNA (*vst-1(gk308047)* genomic DNA digested with EcoRV, injected at 202 ng/μl), and co-injection plasmid (*Punc-122::GFP*, injected at 65 ng/μl). For lines with the *Pvst-1::vst-1::GFP* translational reporter used for Fig. 2 (FQ1137, FQ1149, FQ1167), a fosmid (pJC77) carrying the reporter was amplified by PCR using primers pJC206 and pJC207 to obtain a 9.6 kb product (pJC106). For lines with the *Peat-4::eat-4::mCherry* translational reporter (FQ1167), a fosmid (pJC52) carrying the reporter was amplified by PCR using primers pJC221 and pJC220 to obtain a 14.2 kb product (pJC149). These PCR products were gel-purified and injected at 54 (pJC106) and 44 (pJC149) ng/μl concentrations, respectively, to generate each line.

**Plasmids and primers**. The plasmids, fosmids, and primers used for this study are listed in Supplementary Table 3. The sense and antisense plasmids used for RNAi microinjection were constructed according to Esposito et al.[58]. *vst-1* RNAi plasmids, pJC141 and pJC143, were generated by amplifying *vst-1* cDNA by PCR using JC178/JC179 (sense) and JC180/JC181 (antisense) primer sets, respectively, and targeted 356 nucleotides spanning exons 2–4 of *vst-1* (nucleotides 103–458 of cDNA). *eat-4* RNAi plasmids, pJC133 and pJC135, were generated by amplifying *eat-4* cDNA by PCR using JC170/JC171 (sense) and JC172/JC173 (antisense) primer sets, respectively, and targeted 350 nucleotides spanning exons 3–5 of *eat-4* (nucleotides 211–560 of cDNA). The amplified sense or antisense fragments were placed behind a *flp-17* promoter by Gibson assembly[59]. GFP or mCherry cassettes were inserted into the *vst-1* and *eat-4* fosmids according to Tursun et al.[60].

**RNASeq data analysis**. From the RNA-sequencing (RNASeq) data set in Horowitz et al.[32] (GEO accession: GSE137267; Accession for BAG data set: GSM4074164 and GSM4074165; and Accession for unsorted cells data set: GSM4074162 and GSM4074163), data analysis was performed essentially as described previously[61]. The coverage histograms for *eat-4* and *vst-1* were visualized using the Integrative Genomics Viewer[62] and the mean read counts were quantified using Deeptools[63]. Fold enrichment and false discovery rate were computed using the DESeq2 package[64].

**Sequence alignment**. The phylogenetic tree in Fig. 1d was generated using the multiple sequence alignment program T-Coffee (www.ebi.ac.uk/Tools/msa/tcoffee). Supplementary Fig. 1 shows the result of this alignment using ALINE[65].

**CO$_2$ chemotaxis assays**. Twenty to sixty adult hermaphrodites were washed in M9 solution and placed on unseeded 10 cm Nematode Growth Medium (NGM) plates. A custom-made chamber with a thin layer of glycerol applied to the edges to prevent animals from escaping was pressed into the NGM agar plates. The chamber was 6 cm in diameter and had two gas inlets 2 cm from the center and on opposite sides. Air (20% $O_2$ and $N_2$ balance) flowed into one inlet and $CO_2$ (10% $CO_2$, 20% $O_2$, and $N_2$ balance) flowed into the other at 1.5 ml/min using a dual syringe pump (New Era). The number of worms on each side were counted after 30 min and an avoidance index (AI) was computed by:

AI = (number of worms on air side − number of worms on $CO_2$ side)/(total number of worms). Kruskal–Wallis and Dunn's multiple comparison tests were used for statistical comparisons between wild type (or *vst-1* depending on the experiment) and other genotypes. A few trials were represented in multiple figures, because multiple experiments were simultaneously conducted on a single day: one trial for wild type and two trials for *vst-1* were used for Figs. 1e, 4b, and 6f, and two trials each of wild type, *vst-1*, and *eat-4* were used for both Figs. 1e and 4b. RIA ablation was confirmed before conducting chemotaxis assays for the ablation lines by checking for mCherry expression in RIAs under a fluorescent dissecting microscope.

**Quantification of cellular expression of VST-1 in glutamatergic neurons**. Co-expression of *vst-1* and *eat-4* nuclear reporters was measured by manually labeling each YFP-positive, mCherry-positive, and double-positive nucleus with a unique ID (visible in both YFP and mCherry channels, and on all optical slices) in ImageJ[66], while progressing through the entire head one optical slice at a time.

**Electron microscopy**. *Pvst-1::vst-1::GFP; Peat-4::eat-4::mCherry; Punc-122::GFP* transgenic worms were subject to high-pressure freezing, freeze substitution, embedment in Lowicryl HM20, and postembedding immunolabeling according to Hall et al.[67]. Four to five worms were aligned in one direction into a hat filled with yeast. The hat was frozen under high pressure (2100 bar) and subsequently immersed in liquid nitrogen. Samples were fixed with 0.2% glutaraldehyde/98% acetone/2% water at −90 °C for 110 h, slowly warmed up to −20 °C (5 °C/h), held at −20 °C for 16 h, slowly warmed up to 0 °C (6 °C/h), and held at 0 °C. The samples were rinsed in 100% acetone 4 × 15 min at 0 °C, removed from the hat, placed in HM20 acetone (1 : 2 ratio), and held at 4 °C for 3 h. Solution was changed to HM20 acetone (2 : 1) at 4 °C for 10–20 h, then changed to 100% HM20 five times over 48 h and held at −20 °C. Samples were transferred to gelatin capsules, filled with HM20, sealed, and cured under ultraviolet light for 1–2 days at −20 °C. Ultrathin sections (70 nm) were cut by a Leica UC6/FCS microtome. Immunostaining was conducted using an anti-GFP primary antibody (Millipore Sigma, ab3080, 1 : 10 dilution) and 15 nm Protein A gold-conjugated secondary antibody (PA15, Cell Microscopy Center, University Medical Center Utrecht, 35584 CX Utrecht, The Netherlands, 1 : 50 dilution) on a carbon-formvar-coated copper grid. No primary antibody solution was used for control staining. For co-labeling, the sections were stained with the antibodies mentioned above for VST-1::GFP, fixed with 2% paraformaldehyde, and stained with the anti-red fluorescent protein (RFP) primary antibody (R10367, ThermoFisher, 1 : 20 dilution) and 5 nm Protein A gold-conjugated secondary antibody (PA5, Cell Microscopy Center, University Medical Center Utrecht, 35584 CX Utrecht, The Netherlands, 1 : 50 dilution) for EAT-4::mCherry. Images were acquired using a Philips CM-12 (FEI, Eindhoven, The Netherlands) transmission electron microscope with a charge-coupled device (CCD) camera (Gatan 4k × 2.7k digital camera, Gatan, Inc., Pleasanton, CA).

**Cellular glutamate release assay**. *C. elegans* embryonic cells from the *Pflp-17::iGluSnFR* line were cultured as previously described[37], to obtain BAG neurons expressing iGluSnFR on the cell membrane directed toward the extracellular solution. Immediately prior to an assay, cells were washed ten times with 3 ml control solution to rinse out residual glutamate from the culture medium. A multichannel perfusion system (Automate Scientific) was used to deliver control (145 mM NaCl, 5 mM KCl, 2 mM CaCl$_2$, 1 mM MgCl$_2$, 10 mM HEPES, 10 mM glucose pH 7.2, 335–345 mOsm) and 100 mM KCl (50 mM NaCl, 100 mM KCl, 2 mM CaCl$_2$, 1 mM MgCl$_2$, 10 mM HEPES, 10 mM glucose pH 7.2, 335–345 mOsm) solutions to cells on peanut lectin-coated MatTek dishes. Images were acquired every 100 ms with a ×20 Nikon objective (air, Numerical Aperture 0.8) on a custom-built microscope with a 488 nm excitation light. Excitation and image acquisition were controlled by Live Acquisition software (Till Photonics). Region of interest (ROI) was selected as a 3 × 3 pixel square located at the tip of neurite and mean pixel intensity was used for analysis. A 3 × 3 pixel ROI located outside the cell was used for background subtraction. Matlab was used to generate a plot showing $\Delta F/F$ over time and calculate area-under-the-curve between 10 and 20 s. To compare iGluSnFR expression between cells (Supplementary Fig. 4c), the mean background-subtracted fluorescence prior to KCl application was normalized to excitation light intensity.

**pH measurements via synaptopHluorin**. Dissociated BAG neurons expressing synaptopHluorin were prepared and changes in synaptopHluorin fluorescence in response to KCl, NH$_4$Cl, and MES treatment were acquired as described above for the cellular glutamate release assay with the following differences. Cell cultures were obtained from *Pgcy-33::snb-1::superecliptic pHluorin; Pflp-17::mStrawberry; Punc-122::mCherry* lines. Cells were not subject to extensive washing, to remove residual glutamate prior to the assay. The cell marker mStrawberry was used to identify BAG cell neurites and used to guide search for synaptopHluorin puncta within BAG neurites. Next, 3 × 3 pixel regions expressing synaptopHluorin within BAG neurites were selected for analysis. Multiple puncta were selected from a single neuron if they were not adjacent.

To identify KCl-responsive puncta, we first assessed the response to 100 mM KCl for 5 s. Punctum with baseline fluorescence values close to background levels [mean (0–10 s) ≤ mean + 3 SD of background signal (0–10 s)] were excluded, because $\Delta F/F$ were too noisy and fluctuating. KCl-responsive puncta were chosen by determining whether the maximum $\Delta F/F$ between 10 and 20 s (after passing through a 1 s moving-average filter) was greater than the baseline $\Delta F/F + 3$ SD [max (10–20 s) > mean + 3 SD of baseline signal (0–10 s) using traces passed through a 1 s moving-average filter].

KCl-responsive puncta were further subject to pH measurements as previously described[42,68] with the following details. NH$_4$Cl (95 mM NaCl, 50 mM NH$_4$Cl, 5 mM KCl, 2 mM CaCl$_2$, 1 mM MgCl$_2$, 10 mM HEPES, 10 mM glucose pH 7.2, 335–345 mOsm) was applied for 40 s (control saline 20 s–NH$_4$Cl 40 s–control saline 20 s). MES (145 mM NaCl, 5 mM KCl, 2 mM CaCl$_2$, 1 mM MgCl$_2$, 10 mM MES, 10 mM glucose pH 5.5, 335–345 mOsm) was applied for 60 s (control saline 20 s–MES 60 s–control saline 120 s). Cells were subject to additional washes by control saline (145 mM NaCl, 5 mM KCl, 2 mM CaCl$_2$, 1 mM MgCl$_2$, 10 mM HEPES, 10 mM glucose pH 7.2, 335–345 mOsm) between different treatments to ensure clearing of the previous treatment from the perfusion system.

Fractional increase of synaptopHluorin signal in response to NH$_4$Cl ($\gamma$) for each punctum was acquired by determining the maximum $\Delta F/F$ during NH$_4$Cl application (20–60 s) after passing the trace through a 5 s moving-average filter to remove high-frequency fluctuations hindering accurate measurements. Exclusion criteria used for KCl responders were applied to this experiment to define NH$_4$Cl responders: puncta with baseline fluorescence values close to background levels (within 3 SD) or with a maximum $\Delta F/F \leq 3$ SD above baseline (after passing through a 1 s moving-average filter) were excluded from pH calculation. Fractional decrease of signal in response to MES ($\varepsilon$) was acquired by determining the minimum $\Delta F/F$ during MES application (20–80 s) after passing the trace through the same moving-average filter and applying the same exclusion criteria as for NH$_4$Cl to define MES responders. The pH for each punctum was obtained from measurements of $\gamma$ and $\varepsilon$ using equations described in Mitchell et al.[68] (replaced 7.4 with 7.2 due to difference in pH of NH$_4$Cl solution used):

$$\text{SF} = \frac{\varepsilon}{1 - \frac{1 + 10^{pK-pH}}{1 + 10^{pK-5.5}} + \frac{(1-\varepsilon)\left(10^{pK-pH}-10^{pK-7.2}\right)}{1 + 10^{pK-7.2}}} \tag{1}$$

$$\text{SF} = \frac{\frac{10^{pK-pH}-10^{pK-7.2}}{1 + 10^{pK-7.2}} - \gamma}{\frac{\left(10^{pK-pH}-10^{pK-7.2}\right)(\gamma+1)}{1 + 10^{pK-7.2}}} \tag{2}$$

We used $pK = 7.18$ for superecliptic pHluorin[42] and solved for pH and surface fraction (SF) by finding the intersection of the two functions. Finally, pH values calculated to be below 5.0 were excluded, as they were outside the dynamic range of superecliptic pHluorin[41]. The number of KCl-responsive puncta excluded for pH measurements by these criteria were 6 (of 118), 5 (of 76), and 5 (of 115) for wild type, *vst-1*, and *eat-4*, respectively.

As the vesicular pH was slightly different between genotypes, the maximal pHluorin signal in response to KCl for genotypes with more alkaline vesicles underestimates the extent of vesicle fusion compared to those with more acidic vesicles. In order to account for the effect of baseline vesicular pH on maximal response to KCl, we adjusted the pHluorin signals of the mutants so that it would reflect the increase in pHluorin signal if they had the same baseline vesicular pH as wild type. The fractional increase in pHluorin signal during an exocytosis event when the pH increases from pH$_a$ to 7.2 is proportional to[42]:

$$\delta = \frac{\frac{1}{1+10^{pK-7.2}} - \frac{1}{1+10^{pK-pHa}}}{\frac{1}{1+10^{pK-pHa}} + \frac{SF}{(1-SF)(1+10^{pK-7.2})}} \tag{3}$$

when SF is the surface fraction of the pHluorin probe and we assume that the number of vesicles is not different among genotypes. Therefore, we multiplied the pHluorin signal from each punctum with a factor, $\delta_{\text{wild type}}/\delta_{\text{mutant}}$, so that we would be comparing the pHluorin signal for a mutant if everything else (including surface fraction) stayed the same but the baseline vesicular pH shifted to that of wild type. These adjustment factors were calculated using the mean pH (6.22, 6.33, and 6.20 for wild type, *vst-1*, and *eat-4*, respectively) and mean SF (0.267, 0.310, and 0.239 for wild type, *vst-1*, and *eat-4*, respectively) for each genotype and $pK = 6.18$ (superecliptic pHluorin). $\delta_{\text{wild type}}/\delta_{vst-1} = 1.15$ for adjustment of the KCl response for the *vst-1* mutant and $\delta_{\text{wild type}}/\delta_{vst-1} = 0.98$ for adjustment of the KCl response for the *eat-4* mutant.

**Videotracking and analysis of acute locomotor response to $CO_2$.** Locomotory response to $CO_2$ was acquired and measured essentially as previously described[69]. A custom-built worm tracker was illuminated with red ring lights and the movement of multiple worms was recorded by a CCD camera (Unibrain, Fire-I 785b). Thirty to sixty fed adult hermaphrodites were washed with M9 solution, placed on an unseeded 10 cm NGM plate and were allowed to move within a 6 cm diameter area encircled by a thin layer of glycerol for 5–10 min so that the solution could evaporate. Within 15 min of washing worms, a 6 cm diameter chamber with a gas inlet 1 cm from its edge is pressed onto the NGM to form a seal and alternating flows of air (20% $O_2$, $N_2$ balance) and $CO_2$ (10% $CO_2$, 20% $O_2$, $N_2$ balance) are allowed to enter the chamber at 1.5 ml/min. Videorecording of the movement and engagement of shuttle valve (Neptune Research, SH360T041, Caldwell, NJ) are controlled by a Matlab script (Nikhil Bhatla). Field of view for tracking was set to capture the movement of worms in the half of the chamber closer to the gas inlet.

After the experiment, a Matlab script (Nikhil Bhatla) is used to identify worms by morphological features and calculate locomotory parameters such as instantaneous speed and heading/angle change. In order to exclude non-worm artifacts, we limited our analysis to objects that lasted at least 30 s and had a minimum speed of 1.17 pixels/s (38.15 μm/s). Upon manual inspection of 200 tracks for each genotype, we found these criteria selectively exclude non-worm objects—<5% of excluded objects are worms for all genotypes except for *eat-4* (7%) and *gcy-9* (5.5%). For plots showing change of speed over time, we calculated the mean instantaneous speed of the population of worms at each time point. We defined a high-angle turn as a change of direction equal or >50°/s[53] and calculated the probability that the population of worms at each time point will execute a high-angle turn to generate plots showing turn probability over time. For statistical comparisons, we found the time point ($t_{min}$ for speed and $t_{max}$ for turn probability) at which mean instantaneous speed is minimum (or turn probability is maximum) for each stimulus window, identified tracks spanning $t_{min} - 15$ to $t_{min} + 17$ s (or $t_{max} - 15$ to $t_{max} + 17$ s), calculated the mean instantaneous speed (or turn probability) over that time window for each individual track and conducted Kruskal–Wallis (and Dunn's multiple comparisons) test among genotypes. For the pre-stimulus (P) window, tracks spanning 465–497 s were used for quantification.

**GCaMP imaging in microfluidic devices.** We used microfluidic devices (Micro-Kosmos and custom-built) described in Chronis et al.[70], but modified the solution flow system to be driven by positive pressure to reduce formation of bubbles in the tubing or channels. A custom-built pressure/valve control box was used to deliver filtered air into solution reservoirs, which were connected to the microfluidic device via tubing. The pressure on all channels was 2–5 psi and the pressure in the side channels (channel 1 and 4 in Chronis et al.[70]) were set 0.5–1 psi lower than that of the control and stimulus solution channels (channel 2 and 3), to allow rapid transition (<300 ms) between control and stimulus solutions (Supplementary Fig. 9). Transition between solutions was mediated by a two-way pinch valve (Cole Parmer, GH-98302-02). The stimulus solution was 18.45 mM $NaHCO_3$ in S Basal medium (equilibrates with 10% $CO_2$ and pH 7.2 assuming closed system) and fluorescein (Cole Parmer, #00298-17, 1 : 10,000,000 dilution) was added to visualize the presence of stimulus. Red food dye (McCormick, 930651) was used instead of fluorescein for GCaMP experiments for AVA. NaCl was added to S Basal medium and the pH was adjusted to match the osmolarity and pH of the control solution to that of the stimulus solution. Images were acquired on the inverted microscope as described above using a ×20 objective (Nikon, air, 0.8 Numerical Aperture) and the illumination (488 nm), image acquisition, and transition between solutions were driven by the Live Acquisition software. We performed up to three experiments per worm with at least 1 min between the stimulus presentations to allow GCaMP signals to return to baseline levels. We identified AVA neurons in the *Popt-3::GCaMP6* line based on its anterior-most location. All other lines used for GCaMP imaging were specific for the indicated neuron in Fig. 6 and Supplementary Figs. 6 and 7.

Image registration was conducted using the ImageJ StackReg Plugin ("translation" algorithm) and ROIs were selected around the soma for BAG neurons and around the neurites for interneurons. The background ROI used for background subtraction was selected outside of the worm instead of within the worm due to the severe fluctuation of fluorescence intensities of the latter. We found the maximum $\Delta F/F$ of the former is roughly half of that of the latter and acknowledge the former method does not control for the basal levels of fluorescence in the worm not associated with GCaMP expression (such as autofluorescence), but decided to use it to limit fluctuation of $\Delta F/F$ due to fluctuating levels of background fluorescence. The mean intensity of fluorescence of the ROIs was used to generate plots showing $\Delta F/F$ changes over time using Matlab. Occasionally, we observed huge increases of fluorescence in the absence of stimulation, perhaps correlated with head or body movement, during the recordings. In order to avoid the influence of this $CO_2$-independent activity, we excluded individual traces that had big increases in $\Delta F/F$ during the pre-stimulus period: [max (0–10 s) > mode + 5 SD (0–10 s) using traces passed through a 0.5 s moving-average filter]. The mode of the individual recording was used (instead of the mean) as baseline and the mean SD of all recordings (0–10 s) from a genotype was used (instead of the SD of the individual recording between 0 and 10 s), because a recording with a big increase in fluorescence during the pre-stimulus has

a big mean and a big SD. The number of recordings excluded by this criterion for RIA was 2 (of 44), 2 (of 46), 6 (of 43), 3 (of 43), 4 (of 44), and 7 (of 48) for wild type, *vst-1*, *gcy-9*, *gcy-9 vst-1*, *glr-1*, and *glr-1; vst-1*, respectively. To better represent the variability among the individual responses, we used colormaps ranked in descending order according to the mean $\Delta F/F$ value between 10 and 15 s.

**Reporting summary.** Further information on research design is available in the Nature Research Reporting Summary linked to this article.

## Data availability
The RNASeq data set analyzed in this study is available in the Gene Expression Omnibus (GEO) database (GEO accession: GSE137267; Accession for BAG data set: GSM4074164 and GSM4074165; and Accession for unsorted cells data set: GSM4074162 and GSM4074163). All other data sets generated and/or analyzed during the current study are available from the corresponding author upon request. Source data are provided with this paper.

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

## Acknowledgements
This work was supported by NIGMS grant R35 GM122573 and F31 NS100360 (L.B.H.). Chris Petzold, Kristen Dancel-Manning, and Alice Liang at NYULH DART Microscopy Laboratory, which is partially supported by Laura and Isaac Perlmutter Cancer Center Support Grant NIH/NCI P30CA016087, prepared samples for electron microscopy and acquired electron micrographs. Katie Eyring generated the *Popt-3::GCaMP6f* strain and Elver Ho generated the pEH62 fosmid. Nikhil Bhatla and Alice Fok helped us establish protocols for videotracking. We thank Loren Looger, Jeremy Dittman, Andrew Gordus, Sharad Ramanathan, and Daniel Colon-Ramos for providing plasmids and strains used in this study. We also thank Hang Lu for instruction and advice on microfluidics, Jeremy Dittman for his expertise on pHluorin analysis, and Da-Neng Wang, David Sauer, Nicolas Tritsch, Villu Maricq, and Oliver Hobert for critical discussions. Some strains used in this study were provided by the *Caenorhabditis* Genetics Center, which is supported by the NIH Office of Research Infrastructure Programs grant P40 OD010440.

## Author contributions
J.-H.C. designed and performed all experiments and analyses, except BAG transcriptome studies, which were performed by L.B.H. N.R. contributed to experimental design and data analysis. J.-H.C. and N.R. formatted data for presentation and wrote the manuscript.

## Competing interests
The authors declare no competing interests.
